# A Review of Swine Breeding Herd Biosecurity in the United States to Prevent Virus Entry Using Porcine Reproductive and Respiratory Syndrome Virus as a Model Pathogen

**DOI:** 10.3390/ani14182694

**Published:** 2024-09-16

**Authors:** Satoshi Otake, Mio Yoshida, Scott Dee

**Affiliations:** 1Swine Extension & Consulting, Inc., Shibata 957-0021, Niigata, Japan; mioyoshida@swext-consulting.co.jp; 2Pipestone Research, Pipestone, MN 55482, USA; scottdee7255@gmail.com

**Keywords:** biosecurity, swine, breeding, herd, transmission, prevention, porcine, reproductive, respiratory, syndrome, virus, PRRSV

## Abstract

**Simple Summary:**

Maintaining a sustainable supply of animal protein is the mission of the global swine production industry. The entry of infectious pathogens to swine populations can cause significant animal welfare issues, increase the use of antibiotics, challenge environmental stability, and interrupt/reduce the supply of pork; therefore, preventing pathogen entry is critical to achieve its mission using science-based biosecurity programs. Biosecurity is the application of science-based protocols to minimize the risk of pathogen entry. The objective of this review is to summarize basic biosecurity terms and concepts, review the transmission of porcine reproductive and respiratory syndrome virus (PRRSV) and the biosecurity protocols designed to mitigate these risk factors, and discuss how the swine industry is applying Next Generation Biosecurity to prevent PRRSV infection of the breeding herd.

**Abstract:**

The prevention of disease introduction into swine herds requires the practice of science-based protocols of biosecurity that have been validated to reduce the risk of the entry of targeted pathogens. The fundamental pillars of biosecurity include bio-exclusion, biocontainment, and bio-management. Biosecurity protocols must be science-based, a way of life, continuously validated, cost-effective, and benchmarked over time. This paper will review these concepts, the direct and indirect routes of transmission of porcine reproductive and respiratory syndrome virus (PRRSV), and the interventions that have been designed and validated to prevent infection of the breeding herd. It will close with a review of Next Generation Biosecurity, describing how a science-based approach is being used to prevent PRRSV infection in breeding herds from a large commercial pork production system in the US.

## 1. Introduction

Maintaining a sustainable supply of animal protein to consumers is the mission of the global swine industry. The introduction of infectious pathogens such as porcine reproductive and respiratory syndrome (PRRSV), porcine epidemic diarrhea virus (PEDV), and African swine fever virus (ASFV) to swine herds can disrupt the supply chain, resulting in reduced animal welfare, decreased production of pork, and significant economic loss to farmers. Therefore, preventing the introduction of infectious agents through the practice of biosecurity is critical to achieve this mission. Successful biosecurity requires a culture of confidence throughout the workforce, with all members of the farm team believing in a program of disease prevention, based on the understanding that it has been designed using science-based protocols validated to reduce the risk of targeted pathogen(s) entry.

Regarding difficult pathogens, throughout the global swine industry, no pathogen has challenged biosecurity and negatively impacted animal welfare to a greater degree than porcine reproductive and respiratory syndrome virus (PRRSV). This virus has been estimated to cost the US swine industry USD 660 million/year due to reduced performance, higher mortality, and increased costs of treatment and prevention [1]. In the absence of vaccines capable of inducing sterilizing immunity, the key to controlling PRRS is to prevent PRRSV entry to breeding herds, using biosecurity programs that utilize science-based protocols. Therefore, this review will use PRRSV as a model pathogen that has driven change in the biosecurity practices of the global swine industry.

To begin the discussion, this paper opens with a review of the concepts of biosecurity (Section 1) and then transitions into a focused review of the routes of transmission of PRRSV and the biosecurity protocols validated to reduce these risks at the level of the breeding herd (Section 2). It concludes with a review of how all this information was used to develop a new concept from the United States known as Next Generation Biosecurity (NGB) and the results following its application to a commercial system of pork production to specifically prevent virus entry to breeding herds (Section 3). The authors hope that this paper will provide a resource for farmers and veterinarians to refer to when reviewing biosecurity plans for PRRSV, as well as to instill hope that the prevention of PRRSV reinfection of breeding herds is possible, repeatable, and sustainable.

## 2. A Review of the Concepts of Biosecurity

The definition of biosecurity is “security from transmission or introduction of infectious diseases, parasites, and pests” [2]. This resource defines the three pillars of biosecurity as (1) bio-exclusion, (2) biocontainment, and (3) bio-management.

### 2.1. Bio-Exclusion

Bio-exclusion focuses on preventing the introduction of pathogens to animal populations. In the swine industry, it is routinely applied at the level of the breeding stock supplier, i.e., the artificial insemination center, the genetic nucleus sow farm, and the gilt multiplication unit. However, with the emergence of highly virulent variants of PRRSV, the philosophy of bio-exclusion has also become important at the level of the commercial sow farm, particularly if the herd is in a dense region of swine production.

### 2.2. Biocontainment

The concept of biocontainment has been well established in human clinical medicine and animal laboratory research [3]. As it refers to livestock, biocontainment focuses on internal biosecurity or the practice of biosecurity within the farm. Its goal is to minimize the transmission of pathogens that already exist in the herd population by attempting to prevent their spread within the population, as well as to other farms in the region. The concept of biocontainment is also important in wean-to-market populations, particularly during area regional disease control efforts in high swine dense regions.

### 2.3. Bio-Management

Bio-management focuses on the oversight and execution of the processes and protocols that are associated with a biosecurity program, particularly at the level of the farm personnel. For example, the education and training of the staff/team/organization, along with a regular program of auditing to document compliance, are critical components of a plan of bio-management [4,5].

## 3. What Are the Components of a Successful Biosecurity Program?

The success of the biosecurity program is based on the attitudes of the workforce at the farm and their ability to carry out recommended best management practices for the prevention of pathogen entry to their respective herds [6]. In their excellent review of the role of the farmer in the adoption of biosecurity-based recommendations, Ritter et al. stated that “Approaches that appeal to farmers’ internal motivators or that unconsciously elicit the desired behavior will increase the success of the intervention” [6]. After many years of experience in the swine industry, it is the opinion of the authors that there are five essential components of a successful biosecurity program, all of which are heavily dependent on the belief in, and the acceptance of, a program of disease prevention among all members of the farm workforce. Specifically, a biosecurity program must be:(1)science-based,(2)a way of life,(3)continuously validated,(4)cost-effective,(5)benchmarked over time.

### 3.1. Biosecurity Must Be Science-Based

To prevent the risk of the transmission/introduction of pathogens between/within the units, it is first essential to understand the routes of transmission of the targeted pathogen(s). Once understood, intervention strategies can then be designed to minimize the risk(s). Therefore, all biosecurity protocols should be based on published evidence, as the scientific basis of the program is what helps build the confidence of herd owners and on-farm personnel, thereby driving acceptance and promoting compliance, even during periods of perceived program failure following an unexpected disease outbreak.

### 3.2. Biosecurity Must Be a Way of Life

The practice of biosecurity is challenging, as it requires extra time, continuous attention to detail, and consistent day-to-day application. For example, showering into a breeding herd to reduce the introduction of pathogens via fomites such as personnel clothing and footwear takes time. Filtering incoming air to prevent the airborne entry of pathogens requires a change in human behavior. The effort it takes to clean and disinfect transport vehicles after pig deliveries is physically demanding work. Therefore, a culture that believes that biosecurity is not just a job but is a way of life is an important characteristic of a successful farm. Personnel need to believe in the importance of biosecurity and practice it every day. For this to happen, people need to be educated in the science behind the interventions, be trained to consistently execute them properly, and live the belief that they are effective for preventing disease.

### 3.3. Biosecurity Must Be Continuously Validated

In the operational point of view, it is the daily execution of protocols by people (farm staff, team, organization, etc.) that determines whether a biosecurity program will succeed or fail. As mentioned, this often requires a change in culture on the farm regarding the importance of the program. Once established and implemented, there should be continuous education, the monitoring of compliance, and regular auditing to ensure proper execution. Audits should be unannounced and conducted by swine veterinarians and trained personnel originating from external, third-party sources. Following the audit, follow-up discussions between the farm veterinarian and the staff should take place to maximize learning and minimize future mistakes.

### 3.4. Biosecurity Must Be Cost-Effective

As biosecurity protocols frequently require capital investments, these costs should be calculated and, whenever possible, compared to the cost of the targeted disease(s). This helps all involved, including lenders, to understand the potential return on an investment if following a reduction in the incidence of the targeted disease(s).

### 3.5. Biosecurity Must Be Benchmarked over Time

Benchmarking the outcomes from a biosecurity program over time, be it at the population, herd, site, regional, or national level, is important to measure success and/or failure and to drive continuous improvement. One example of a useful benchmarking metric is the measurement of the percent incidence risk, i.e., the total number of new pathogen introductions divided by the number of herds observed. An excellent example of a system that uses this approach to measure incidence risk is the University of Minnesota Dr. Bob Morrison Swine Health Monitoring Project (MSHMP) [7]. This approach has been applied to multiple agents, including PRRSV, PEDV, and *Mycoplasma hyopneumoniae*. Also, the use of software-generated maps that display the location and pathogen status of neighboring herds in an area is particularly helpful for monitoring the progress of regional disease control programs.

## 4. Transmission of Porcine Reproductive and Respiratory Syndrome Virus and the Biosecurity Protocols Designed to Prevent Infection of the Breeding Herd

The disease of porcine reproductive and respiratory syndrome (PRRS) has been a driver of change throughout the global swine industry, especially regarding the practice of biosecurity at the level of breeding herds. PRRS is caused by PRRSV, a single-stranded enveloped RNA virus, classified in the genus *Arterivirus* and a member of family *Arteriviridae*, which was first identified in 1991 [8,9]. PRRSV undergoes genetic change that results in the regular emergence of new variants that often display enhanced pathogenicity and improved ability to spread [10,11]. Recent variants demonstrating these features include PRRSV 184, PRRSV 174, and PRRSV 144 lineage L1C [12,13]. For example, these variants have adapted efficient means to transmit within and between herds, including an improved ability to spread via the airborne route [14]. PRRSV is now known to be endemic in pig populations across five continents, and only African swine fever virus can rival its clinical and economic significance throughout the global swine industry [15]. The emergence of PRRSV, along with its continued adaptation and economic significance, has forced swine producers and veterinarians to focus intensively on preventing viral entry to breeding herds using science-based biosecurity. However, despite years of effort, a long-established paradigm has been that, while the elimination of PRRSV from infected herds is likely to be successful, the prevention of reinfection with a new variant is not. For the global swine industry to gain control of PRRS, this paradigm must change, and the first requirement to make this change is developing a comprehensive understanding of the routes of the transmission of PRRSV and the biosecurity protocols that have been proven to mitigate these risks and prevent infection. The paper will now review the published direct and indirect routes of PRRSV, reference the biosecurity protocols that have been validated to mitigate these risks, and describe industry actions that have been put into place at the level of the breeding herd.

### 4.1. Pigs

Pig-to-pig transmission is the principal means of transmission of many infectious pathogens of swine, including PRRSV. PRRSV produces a persistent infection and can survive in the tonsils and lymph nodes of infected pigs for extended periods, i.e., 157 days post-infection. In addition, shedding of the virus from carrier animals, resulting in the infection of naïve sentinels, can occur for up to 49–86 days [16,17]. Based on this information, it is essential that producers purchase breeding stock from a known PRRSV-negative source and quarantine/test these animals prior to their entry into the breeding herd.

### 4.2. Semen

PRRSV infection of breeding females can occur following insemination with PRRSV-positive semen [18]. The frequency of PRRSV shedding in semen is highest during the acute phases of the infection; however, intermittent shedding can occur days to months after infection. Swenson et al. detected viable virus in the semen from experimentally infected boars up to 43 days, and Christopher-Hennings et al. detected viral RNA by PCR in the semen from experimentally infected boars up to 92 days following inoculation [19,20]. Furthermore, Gradil et al. also demonstrated the transmission of PRRSV via extended semen from experimentally infected boars [21]. Therefore, as with breeding stock, only semen from PRRSV-negative boar populations should be used, and in the artificial insemination center, boars who provided semen should be blood tested on the day of collection, followed by PCR testing to document the absence of the virus, prior to shipment to breeding herds.

### 4.3. Needles

Needles are an efficient fomite for the transmission of swine pathogens, particularly during the viremic period of infection. In commercial swine farms, pigs receive numerous intramuscular injections of vaccines and antibiotics, and it is not a common practice to change needles between individual pigs, due to cost and labor constraints. It is well documented that needles can serve as mechanical vectors of PRRSV due to the prolonged viremia that occurs following infection. Under experimental conditions, Otake et al. demonstrated the transmission of PRRSV from viremic infected pigs to naïve pigs in the absence of a needle change between animals [22]. Furthermore, Madapong et al. reported that the use of a “needle-free” syringe prevented the hematogenous transfer of PRRSV between viremic and susceptible pigs, while sentinel pigs became infected following the injection of viremic pigs in the absence of a needle change [23]. With this in mind, needle-free devices should be considered for use in breeding herds and/or needles should be changed between individual sows, particularly those in the third trimester of gestation.

### 4.4. Fomites

Fomites such as coveralls, boots, or other inanimate objectives have also been considered as risk factors in the transmission of swine pathogens. Pirtle and Beran assessed the survivability of PRRSV in/on 16 fomites, including plastic, stainless steel, rubber, alfalfa, wood shavings, straw, corn, swine starter feed, denim cloth, phosphate-buffered saline, saline G, well water, city water, swine saliva, urine, and fecal slurry [24]. An attempt was done to isolate viable PRRSV from these samples after storage at 25 to 27° C. Viable PRRSV was isolated from phosphate-buffered saline through day three, saline G through day six, well water through day eight, and city water through day eleven following the inoculation. In contrast, viable PRRSV was not recovered from any other fomite samples beyond 30 min post-inoculation. Using a model involving a coordinated sequence of events routinely experienced in a “day in the life” of a pork producer, Dee et al. demonstrated that viable PRRSV could be mechanically transported from a truck wash environment onto the floormats of transport vehicles and the soles of personnel footwear and walked into a mock farm, resulting in the contamination of incoming farm supplies and the hands of personnel in 8 out of 10 replicates during cold weather and 2 out of 10 replicates during warm weather [25,26]. Other materials such as Styrofoam, cardboard, linoleum, concrete, and stainless steel have also been reported to harbor viable PRRSV for different amounts of time [25,27]. Dee et al. also indicated that the use of disposable boots, bleach boot baths, or bag-in-a-box shipping methods was highly efficacious in preventing the mechanical transmission of PRRSV [28]. Based on the risk of fomites such as incoming supplies, commercial farms have installed D&D (Disinfection and Downtime) rooms, a heated facility with a separate airspace from the farm facility proper. Incoming supplies are delivered to the D&D room from a commercial vendor to a designated employee of the farm, i.e., the manager, who receives the supplies, applies disinfectant to the exterior of the containers housing the supplies, and allows all materials to remain in the locked room for a designated period of time before entering the farm.

### 4.5. Personnel

Personnel movement within or between production units has long been considered as a risk factor for the transmission of swine pathogens; therefore, biosecurity protocols that control and regulate personnel entry to farms and movement between farms via downtime policies are frequently practiced. Under experimental conditions, Otake et al. documented that PRRSV could be transmitted to susceptible pigs by coveralls, boots, and hands of personnel that had been previously contaminated following contact with infected pigs. The study also demonstrated that changing coveralls and boots, along with washing hands or use of a shower in/shower out protocol between infected and susceptible groups prevented transmission of the virus [29]. Pitkin et al. also reported, under the conditions of the study, PRRSV RNA was readily detected on several different fomites and on the hands of personnel after contact with infected swine [30]. In addition, the transmission of PRRSV to sentinel pigs was observed in seven out of seven replicates in which biosecurity practices such as changing clothing and footwear or washing hands were not practiced. In contrast, when these basic practices were followed, transmission was prevented every time (0/7 replicates). Furthermore, Pitkin demonstrated that, following contact with groups of pigs that had been experimentally inoculated with PRRSV and *Mycoplasma hyopneumoniae,* the combination of a shower in/out protocol and a one-night downtime period prevented the spread of both pathogens to susceptible pigs [31]. Therefore, the use of personnel downtime prior to entry to breeding herds, in combination with a shower in/shower out protocol prior to entering the animal airspace, are common practices across modern swine production companies in the US. Another practice that has been devised based on this information is the “boot bench”, a wooden bench that divides the farm anteroom into “clean” and “dirty” segments. The goal of the boot bench is to minimize the chance of the soles of personnel footwear to spread pathogens throughout the anteroom. Upon entry to the anteroom from the outside environment, personnel sit on the bench, remove their footwear, and place it on the floor of the “dirty” side of the bench, elevate their feet and rotate 180°, place their feet on the opposite (“clean”) side of the bench, and walk to the shower facility.

### 4.6. Transport

Transport vehicles such as livestock trailers, feed trucks, and motorized vehicles used by farmworkers or veterinarians have been postulated to play a significant role in the transmission of swine pathogens. It is logical to assume that organisms can be carried on the frame of vehicles or in manure embedded in tire treads. The studies conducted by Dee et al. demonstrated that contaminated vehicles played a role in the transmission of PRRSV under controlled conditions depicting a “coordinated sequence of events” that swine farmers routinely practice under differing environmental conditions [25,26]. Both studies demonstrated how transport vehicles can act as fomites for the long-distance (50 km) spread of PRRSV. Dee et al. also demonstrated that pigs housed in model trailers intentionally contaminated with PRRSV could become infected [32]. This study involved housing experimentally PRRSV-infected seeder pigs in model trailers for two hours, followed by their removal and the introduction of naïve sentinel pigs in the absence of sanitation of the trailer model. Infection of the sentinels was demonstrated in three out of four replicates. The result concluded that transport vehicle sanitation is an important strategy to prevent the transmission of PRRSV to susceptible pigs. Galvis et al. developed and calibrated a mathematical model for the transmission of PRRSV, finding that 36% to 80% of PRRSV infections could be associated with the transport of animals between sites [33]. These collective observations provide evidence to support the importance of the risk of transport for spread of the virus. It is well accepted that transport vehicle sanitation is critical to mitigate risk. Therefore, in today’s commercial industry, a program of transport sanitation, including the removal of organic material, followed by washing, the application of a validated product to disinfect the trailer, and allowing it to thoroughly dry, either through the short-term application of heated air (70 °C for 30 min) or the circulation of room temperature air (20 °C) for an overnight period, are common practices across commercial companies [34,35,36].

### 4.7. Avian Species

Birds have often been proposed as a risk factor for the transmission of swine pathogens between farms due to their ability to travel great distances. Direct transmission of pathogens from wild birds to pigs can occur through mechanical means, particularly in backyard farms where interactions between birds and pigs can occur. Zimmerman et al. reported that mallard ducks orally exposed to PRRSV shed the virus in feces. [37]. In addition, pigs intranasally exposed to PRRSV isolated from mallard feces became viremic, seroconverted, and transmitted the virus to sentinel swine. However, Trincado et al. attempted to reevaluate the potential of avian species in the transmission of PRRSV and concluded that mallard ducks were not likely to serve as reservoirs of PRRSV [38,39]. In this study, adult mallard ducks were exposed to PRRSV in the drinking water, as well as orally inoculated with PRRSV. However, the infection of PRRSV did not occur in the ducks, as PRRSV RNA was not detected in duck feces and naïve sentinel pigs exposed to the feces did not become infected. Additionally, PRRSV-infected pigs did not transmit the virus to mallard ducks following 21 days of direct contact under controlled conditions. Therefore, based on this contradictory information, the risk of PRRSV transmission by avian species is not currently considered of high importance.

### 4.8. Rodents

Based on published data, rodents do not appear to be vectors of PRRSV. Hooper et al. documented the inability to recover PRRSV from several tissues (sera, lung, thymus, and spleen) of mice and rats trapped on a swine farm with endemic PRRSV infection [40]. Furthermore, mice and rats experimentally inoculated with PRRSV did not become infected [40]. Wills et al. also examined the susceptibility of rats and mice for PRRSV and reported similar results [41]. However, despite these negative results, an on-farm rodent control program is very important for the control of other diseases, as well as to minimize the damage that rodents can cause to a swine facility.

### 4.9. Feral/Domestic Animals

In addition to rodents, feral and domestic animals have been hypothesized to be vectors of swine pathogens, especially as it pertains to wild pigs, which have been postulated to serve as disease reservoirs. In contrast to the well-documented risk of wild pigs for the spread of ASFV, serological surveys of wild pigs have demonstrated a low prevalence of PRRSV antibodies [42,43,44]. Regarding the risk of non-swine species as risk factors for PRRSV spread, Wills et al. inoculated dogs, cats, skunks, opossums, and racoons with PRRSV via the intranasal and intramuscular routes, and serum samples were collected over the course of 21 days post-inoculation, and tonsils and lymph nodes were collected upon euthanasia 21 days post-inoculation [41]. All samples were negative by PCR and virus isolation, except for two serum samples from two out of four opossums that were PCR-positive on days 3 and 14 post-inoculation. However, due to their weak Ct values, these values were concluded to be false positives; therefore, non-porcine mammalian species do not appear to serve as reservoirs of PRRSV.

### 4.10. Insects

It is well documented that various species of insects can serve as vectors of swine pathogens, including ASFV, Japanese B encephalitis virus, and vesicular stomatitis virus [45,46,47]. Regarding PRRSV, Otake et al. demonstrated PRRSV transmission to susceptible pigs following contact with houseflies (*Musca domestica*) allowed to feed on viremic pigs [48]. Otake et al. also showed the survivability of infectious PRRSV in the midgut of houseflies for up to 6 h [49]. Furthermore, Otake et al. demonstrated that 2 out of 10 individual houseflies, which had fed on an experimentally inoculated pig in the acute phase of viremia, could transmit PRRSV to naïve sentinels [50]. These studies collectively concluded that PRRSV can be mechanically transmitted from infected pigs to susceptible pigs by houseflies and PRRSV can survive within housefly populations for up to six hours post-feeding, supporting the hypothesis that house flies could contribute to the horizontal transmission of PRRSV among pigs within infected commercial farms. In addition, Schurrer et al. reported that flies could become contaminated with PRRSV from infected pigs and transport the virus out to 1.7 km from the population [51]. Furthermore, mosquitoes (*Aedes vexans*) have also been hypothesized to be mechanical vectors of PRRSV. Under experimental conditions, Otake et al. demonstrated PRRSV transmission to susceptible pigs via mosquitoes that had previously fed on viremic infected pigs [52]; however, it was determined that the virus was not capable of replicating in the salivary glands of the mosquitoes [53]. These findings indicate that mosquitoes may pose an ability to mechanically transmit PRRSV; however, they do not serve as biological vectors. Based on these collective observations, insect control programs on swine facilities involving the strategic use of insecticides and habitat management are important to minimize the PRRSV risk, maximize animal and personnel comfort, and reduce carcass damage at slaughter secondary to insect bites, particularly in tropical areas of the world where insects are prevalent year-round.

### 4.11. Swine Slurry and Lagoon Effluent

Despite limited research, it has been hypothesized that slurry or lagoon effluent could serve as risk factors for the transmission of PRRSV. Dee et al. reported that samples of lagoon effluent inoculated with PRRSV and stored at 4 °C and 20 °C were PCR-positive up to 12 days post-inoculation [54]. Exposure to PRRSV-contaminated swine effluent stored at 4 °C for up to eight days post-inoculation resulted in infection and seroconversion in pigs. At 20 °C storage, exposure to effluent 1–3 days post-inoculation resulted in PRRSV infection in pigs. These results indicated that (1) the viability of PRRSV in swine effluent is relatively short (1–8 days), (2) its survival in effluent appears to be dependent on temperature, and (3) pigs can become infected following contact with PRRSV-contaminated effluent. In conclusion, the risk of swine herds becoming infected with PRRSV following contact with contaminated lagoon effluent is a time- and temperature-sensitive risk. Recently, Linhares et al. assessed the infectivity of PRRSV in slurry at different temperatures [55]. In this study, PRRSV consistently displayed shorter half-lives in slurry compared to minimum essential media (MEM). Specifically, the PRRSV half-life in MEM and slurry was estimated at 112.6 and 120.5 h at 4 °C, 14.6 and 24.5 h at 20 °C, 1.6 and 1.7 h at 40 °C, 2.9 and 8.5 min at 60 °C, and 0.36–0.59 min at 80 °C, respectively. Therefore, based on these findings, along with the repeatable observation that PRRS outbreaks occur during pit pumping season, special care should be taken to avoid the mechanical risk of the virus spread in contaminated trucks and pumping equipment moving between sites.

### 4.12. Pig Meat

While the ability to transmit PRRSV in pig meat has been evaluated, limited data exist to support this hypothesis. PRRSV from meat samples of experimentally infected pigs was detected for up to 18 days stored at 4 °C and up to 60 days at −20 °C [56]. Furthermore, viable PRRSV was detectable at concentrations ranging from 10^3.3^ to 10^4.3^ TCID_50_/g of muscle tissue from experimentally PRRSV-infected pigs at day 11 post-inoculation, and susceptible pigs consuming these samples became infected [57]. Bloemraad et al. reported low PRRSV loads in muscle tissue collected from viremic pigs, and that viral titer remained stable following storage for up to 48 h at 4 °C [58]. Mager et al. isolated PRRSV from muscle samples collected from the experimentally inoculated pigs at 7 and 14 days post-inoculation; however, they did not detect the virus from muscle samples collected from 44 carcasses at slaughter [59]. In conclusion, the presence of the virus in pig meat from slaughter age animals appears to be very low. In contrast, viable PRRSV has been recovered from meat juice during the viremic period and from human hands after handling the meat, and the mechanical transmission of PRRSV from contaminated hands to susceptible pigs was documented [60]. Therefore, based on this collective information, the entry of pork products to commercial swine farms should be prohibited.

### 4.13. Aerosols

In the initial outbreaks of PRRSV in Europe, widespread transmission of the virus between herds in affected regions was initially attributed to airborne spread; however, experimental reproduction of the aerosol transmission of PRRSV was initially difficult to achieve and was documented only over short distances (1 m) [61,62]. At this time, the role of aerosols in the transmission of PRRSV has been well documented. With the emergence of new variants, i.e., PRRSV 184, Dee et al. demonstrated the infection of susceptible animals by aerosolized PRRSV over 150 m using a straight tube model at 27 km/h air speed [63]. Following this information, Otake et al. reported the airborne transport of a mixed infection of PRRSV 184 and *Mycoplasma hyopneumoniae* out to 9.2 km under controlled field conditions [64]. This work led to the investigation of whether the filtration of incoming air could reduce the risk of viral entry to herds. Dee et al. used a production region model to simulate a dense region of swine farming and demonstrated that air filtration systems could successfully prevent the airborne transmission of PRRSV and *Mycoplasma hyopneumoniae* [65]. Based on this information, the filtration of incoming air to breeding herds has been significantly advanced and is a critical component of biosecurity programs for breeding herds [66]. Under commercial field conditions, buildings are ventilated either by positive or negative pressure and utilize commercial filters with fractional efficiencies ranging primarily from MERV 14 to 16. The impact of this intervention has been significant, especially as it pertains to preventing PRRSV aerosol transmission between commercial breeding herds [67].

### 4.14. Feed

In 2014, Dee et al. first described the ability of feed to transmit PEDV to susceptible pigs [68]. Since that time, this hypothesis has been validated by several investigators across multiple viruses, such as Seneca virus A, PRRSV, Classical swine fever virus, Pseudorabies virus, and ASFV [69,70,71]. These studies evaluated the survival of important viral pathogens of livestock in animal feed ingredients imported daily into the United States under simulated transboundary conditions or during long-distance transport across the Continental US. In the transboundary model, PRRSV was recovered from soybean meal 37 days post-inoculation [71]. The experimental transmission of PRRSV through natural feeding behavior has been demonstrated as well [72,73]. In this study, tons of complete feed were contaminated with PRRSV, PEDV, and Seneca virus A using an “ice block challenge” model, which pigs were allowed to consume ad libitum. Finally, Dee et al. documented the transmission of PRRSV in domestic pigs under field conditions following the oral ingestion of feed material, followed by a proof-of-concept study under laboratory conditions [72]. This recent work has led to the regular use of validated feed additives having antiviral properties and being capable of neutralizing PRRSV in feed [72].

## 5. Next Generation Biosecurity: A Model for the Swine Industry

To advance the control of PRRS, a recent publication challenged US swine veterinarians to help farmers to eliminate PRRSV from breeding herds and prevent reinfection through improvements in biosecurity [74]. It had been historically demonstrated that PRRSV could be eliminated from infected breeding herds, resulting in the availability of PRRSV-naïve genetic sources, thereby mitigating viral entry via direct routes of transmission, i.e., infected breeding stock and contaminated semen [75,76,77,78]. However, the reinfection of naïve breeding herds with new variants of the virus despite the use of naïve replacement animals and semen was a frequent event (area spread), particularly when the herds were in regions of dense swine production [79,80]. To build confidence throughout the US industry, it was important to prove that the root cause(s) of area spread could be identified and that reinfection could be prevented/reduced. To solve this problem, a research plan was designed to identify potential indirect routes of PRRSV transmission, including mechanical (fomite-based) routes such as contaminated transport, personnel clothing and footwear and incoming supplies, PRRSV-positive aerosols, and contaminated feed, and to test and validate biosecurity protocols to mitigate these risks. Table 1 provides a timeline summarizing the history of area spread, the sequential milestones of the research plan, and the implementation of a comprehensive, science-based approach to biosecurity, known as Next Generation Biosecurity (NGB) [66].

The concept of NGB was based on the application of validated mitigation strategies across “four pillars” of routes PRRSV entry to a breeding herd, including pillar 1: direct routes (infected pigs and contaminated semen); pillar 2: mechanical routes (transport, personnel, and supplies); pillar 3: the aerosol route; and pillar 4: the route of contaminated feed, depicted in Figure 1 [66]. NGB initially focused on the quarantining and testing incoming genetic stock originating from a documented PRRSV-naïve genetic source in combination with the use of semen from a PRRSV-naïve artificial insemination center. As new information became available from the research plan, NGB was then expanded to include protocols to manage mechanical/fomite-based risks of PRRSV, such as contaminated transport, personnel entry, boots and coveralls, and contaminated supplies. This was followed by the application of validated air filtration protocols to control airborne risk and, later, the use of additives capable of inactivating PRRSV in breeding herd diets. Through the application of this comprehensive, science-based approach, PRRSV incidence risk was documented at the incidence of new PRRSV variant infections across all participating herds at < 10% for 2 years (8.6% year 1 and 9.2% year 2) across a system of pig production that included 76 breeding herds and 381,404 sows, despite a historical incidence risk ranging as high as 40–55% per year.

## 6. Conclusions

In closing, while no other pathogen has challenged the biosecurity protocols of swine breeding herds as aggressively as PRRSV, this review indicated that the literature regarding the routes of transmission is comprehensive and complete, and there is evidence that improvements in biosecurity can successfully lower the PRRSV incidence risk across large-scale commercial production for a period. In support of this conclusion, this topic of Next Generation Biosecurity will be revisited later in this Special Issue on Biosecuring Animal Populations, and follow-up data on the PRRSV incidence risk from the third year, as well as a cumulative summary of the incidence risk over all three years of the project will be provided. In addition, the impact of different levels of biosecurity on the PRRSV incidence risk and breeding herd productivity will be discussed. Clearly, progress is being made in preventing PRRSV infection of breeding herds, with the preliminary efforts also being put forward to biosecure wean-to-market herds. For the first time since its emergence, the paradigm is changing from “PRRSV is controlling the swine industry” to “the swine industry is controlling PRRSV”, which is clearly a welcome change. We do acknowledge that an inherent limitation of this review is that it is “US-centric” and does not explore biosecurity approaches practiced in other countries. This is an excellent topic for a future publication, and perhaps reading this review will stimulate some activity from others.

Finally, it is the authors’ opinions that these are exciting times when the concept of biosecurity has never been more relevant, not only regarding the domestic and transboundary diseases of pigs but to other species groups as well, such as the poultry and dairy industries, which are currently afflicted with highly pathogenic avian influenza virus. Therefore, as viruses will always be drivers of change in animal agriculture, it is time to band together, work as a team, and share our respective knowledge to protect the health and welfare of our global herds and flocks for the benefit of farmers and ranchers.

## Figures and Tables

**Figure 1 animals-14-02694-f001:**
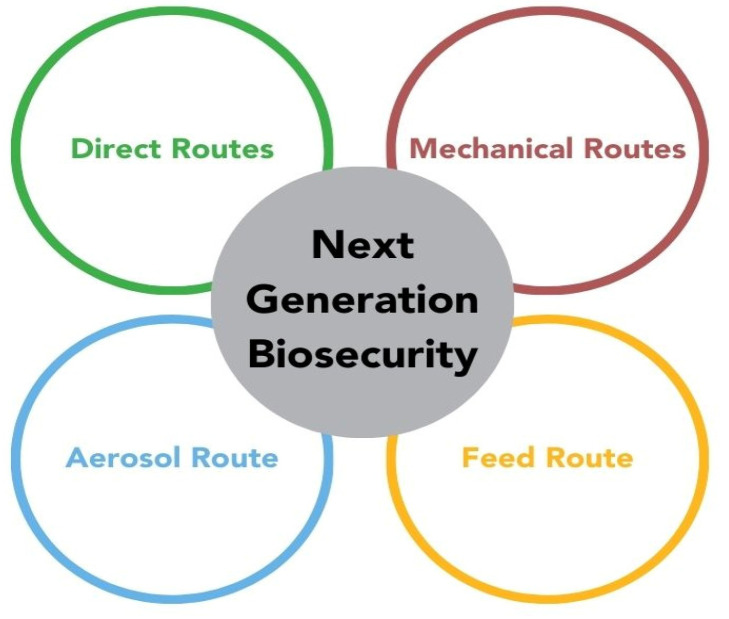
A graphic description of the Next Generation Biosecurity concept and its four pillars. Figure courtesy of Dr. Dee.

**Table 1 animals-14-02694-t001:** Timeline summarizing significant events and key research milestones that led to the development and validation of the Next Generation Biosecurity concept.

Years	Milestone
2000	PRRSV-naïve breeding stock and semen become commercially available.
2001	Area Spread of PRRSV is described.
2002–2004	Mechanical risks (contaminated transport, clothing/footwear, and incoming supplies) for the spread of PRRSV are identified.
2005–2007	Protocols of transport sanitation, personnel entry, and supply entry are validated.
2007–2010	Aerosol transmission and long-distance airborne transport of PRRSV are identified.
2009–2012	Air filtration for PRRSV prevention is validated.
2014–2023	The risk of contaminated feed for the transmission of viruses of veterinary significance is discovered.
2014–2022	Feed biosecurity protocols (additives, storage) are developed.
2021–2024	Next Generation Biosecurity concept developed and tested.

## Data Availability

All data are included in the paper.

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
