# Peer review of "A Review of Swine Breeding Herd Biosecurity in the United States to Prevent Virus Entry Using Porcine Reproductive and Respiratory Syndrome Virus as a Model Pathogen"

_animals, 2024, doi:10.3390/ani14182694_

Round 1

Reviewer 1 Report

Comments and Suggestions for Authors

The manuscript presents a comprehensive but concise review of swine biosecurity protocols relating to the control of PRRS in breeding herds.

In section 3 there is very considerable overlap with the content of reference 63, such that it is not clear what additional material this section presents. This section could therefore be removed, in favour of a slight expansion of the introduction to include some of this content.

Sections 1 and 2 both collate thorough summaries of the available publications, providing a clear and well presented synthesis of each topic, consistently supported by relevant references.

The conclusions are clear, and supported by the review contents.

Comments on the Quality of English Language

Editing is required throughout to correct errors in grammar and spelling.

Author Response

MS 3154156  Response to Reviewers

Greetings. I sincerely would like to thank the review team for helping us to improve our

manuscript. We are excited about this manuscript as it is planned to be the lead off article in

the final publishing of the Special Issue on Biosecuring Animal Populations".

We have attempted to address the points indicated by all three reviewers, and the changes we

made are highlighted in red in the revised manuscript. Specifically, to each point:

Reviewer #1: Comments and Suggestions for Authors

The manuscript presents a comprehensive but concise review of swine biosecurity protocols

relating to the control of PRRS in breeding herds.

THANK YOU VERY MUCH. YES, THAT WAS WHAT WE AIMED FOR.

In section 3 there is very considerable overlap with the content of reference 66, such that it is

not clear what additional material this section presents. This section could therefore be

removed, in favor of a slight expansion of the introduction to include some of this content.

THANK YOU FOR RAISING THIS CONCERN. BASED ON THE OTHER REVIEWERS

REQUEST FOR MORE INFORMATION, WE HAVE EXPANDED SECTION 3 AND

INSERTED A TABLE AND A FIGURE TO BETTER EXPLAIN THE HISTORY AND THE

PRINCIPLES OF THE NEXT GENERATION BIOSECURITY CONCEPT. HOPEFULLY

THIS WILL ENHANCE THE VALUE OF THIS SECTION.

Sections 1 and 2 both collate thorough summaries of the available publications, providing a

clear and well presented synthesis of each topic, consistently supported by relevant references.

The conclusions are clear, and supported by the review contents.

THAK YOU. WE APPRECIATED YOUR COMMENTS.

Reviewer 2 Report

Comments and Suggestions for Authors

The topic of the paper is certainly of considerable interest to the pig industry as PRRS causes high animal losses and economic losses. All in all, the paper proposes in the title a revision of the biosecurity model which then does not appear to be structured in a concrete way in the text. The first section on biosecurity cocnentto as structured is not functional and it would have been better to decline the biosecurity control factors into the 3 pillars reported. Section 2 gives a list of the variables that need to be controlled for PRRS control but without contextualization considering section 1. Section 3 which could be the one of most interest and novelty is instead very minimal. In addition, no specifics were made between avoiding the introduction of the virus and controlling the spread within the herd in the case of an outbreak. 

The research aims to summarize good safety management practices for controlling PRRS The theme is certainly interesting because PRRS is a viral disease present in both American and European pig farms that causes significant losses of livestock and economic. There are no fully effective vaccines so that controlling the triggers is of considerable interest. Works on the same subject are listed in the bibliography. The paper aimed to summarize existing biosafety standards and new standards in an updated manner and this can be an added value. The authors should reformulate the structure according to the indications given. They should also propose a systematic revision of the bibliography and better address the issues. I suggest that the review of factors should be carried out by assessing the effect both in the American and European disputes. The conclusions relate only to the US context. The authors also propose to address the theme of Next Generation Biosecurity, which is only marginally done. The conclusions are not satisfactory. The references can be extended and sometimes systematized Tables and figures are not present. it is suggested to include a table on the articles analysed or a descriptive figure of the points of the company that can be checked.

Comments on the Quality of English Language

The text is quite fluent and I think only minor revision is needed.

Author Response

MS 3154156  Response to Reviewers

Greetings. I sincerely would like to thank the review team for helping us to improve our

manuscript. We are excited about this manuscript as it is planned to be the lead off article in

the final publishing of the Special Issue on Biosecuring Animal Populations".

We have attempted to address the points indicated by all three reviewers, and the changes we

made are highlighted in red in the revised manuscript. Specifically, to each point:

Reviewer #2: Comments and Suggestions for Authors

The topic of the paper is certainly of considerable interest to the pig industry as PRRS causes

high animal losses and economic losses. All in all, the paper proposes in the title a revision of the

biosecurity model which then does not appear to be structured in a concrete way in the text.

THANK YOU FOR THIS IMPORTANT CRITIQUE REGARDING THE LACK OF

INFORMATION ON THE NEXT GENERATION BIOSECURITY MODEL. WE TOOK

YOUR POINT SERIOUSLY, AND EXPANDED THE TEXT IN SECTION 3 AS WELL AS

ADDED A TABLE AND A FIGURE TO BETTER EXPLAIN THE HISTORY OF THE

CONCEPT AND THE PILLARS THAT IT IS BUILT UPON. HOPEFULLY YOU WILL FIND

THESE CHANGES SATISFACTORY.

The first section on biosecurity cocnentto as structured is not functional and it would have been

better to decline the biosecurity control factors into the 3 pillars reported. Section 2 gives a list of

the variables that need to be controlled for PRRS control but without contextualization

considering section 1. Section 3 which could be the one of most interest and novelty is instead

very minimal.

THANK YOU. WE STRUCTURED THIS REVIEW TO INCLUDE 3 SPECIFIC SECTIONS.

THE PURPOSE OF SECTION 1 WAS TO PROVIDE A BROAD OVERVIEW AND

INTRODUCTION/REVIEW INTO SOME BASIC BIOSECURITY TERMINOLOGY. THE

GOAL OF SECTION 2 WAS TO PROVIDE A COMPREHENSIVE REVIEW OF PRRSV RISK

FACTORS AND THE BIOSECURITY PROTOCOLS THAT WERE DEVELOPED AND

VALIDATED TO PREVENT VIRAL ENTRY INTO BREEDING HERDS. THIS LED INTO

SECTION 3 WHICH BASICALLY PUT ALL OF THIS EARLIER INFORMATION

TOGETHER AND INTRODUCED THE NEXT GENERATION BIOSECURITY MODEL.

THIS APPROACH IS EXPLAINED IN THE SIMPLE SUMMARY, THE ABSTRACT AND

ON LINES 52-61 FOLLOWING THE INCLUSION SUPPORTING TEXT.

In addition, no specifics were made between avoiding the introduction of the virus and

controlling the spread within the herd in the case of an outbreak.

WE APOLOGIZE THAT THIS REVIEW IS ONLY FOCUSED ON PREVENTING VIRAL

ENTRY. YOUR IDEA OF CONTROLLING WITHIN THE HERD DURING AN OUTBREAK

IS EXCELLENT, BUT IS BEYOND THE SCOPE OF THIS REVIEW.

The research aims to summarize good safety management practices for controlling PRRS The

theme is certainly interesting because PRRS is a viral disease present in both American and

European pig farms that causes significant losses of livestock and economic. There are no fully

effective vaccines so that controlling the triggers is of considerable interest. Works on the same

subject are listed in the bibliography. The paper aimed to summarize existing biosafety standards

and new standards in an updated manner and this can be an added value. The authors should

reformulate the structure according to the indications given. They should also propose a

systematic revision of the bibliography and better address the issues. I suggest that the review of

factors should be carried out by assessing the effect both in the American and European disputes.

The conclusions relate only to the US context. The authors also propose to address the theme of

Next Generation Biosecurity, which is only marginally done. The conclusions are not

satisfactory. The references can be extended and sometimes systematized Tables and figures are

not present. it is suggested to include a table on the articles analysed or a descriptive figure of the

points of the company that can be checked.

THANK YOU VERY MUCH. HOPEFULLY, OUR REVISIONS TO SECTION 3 (MORE

DETAIL, AS WELL AS A TABLE AND A FIGURE) ARE SUFFICIENT. TO YOUR POINT

ON BIOSECURITY IN EUROPE, THIS IS AN EXCELLENT IDEA AS WELL, BUT IS

BEYOND THE SCOPE OF THE REVIEW. TO MAKE THIS CLEAR FROM THE

BEGINNING, WE CHANGED THE TITLE AND INSERTED CLARIFYING TEXT

REGARDING ITS US-FOCUS AND STATING THIS ACKNOWLEDGED LIMITATION.

PLEASE SEE LINES 55-58 AND 526-529.

Comments on the Quality of English Language: The text is quite fluent and I think only minor

revision is needed.

THANK YOU AGAIN FOR THE TIME YOU SPENT DURING YOUR BUSY DAY TO

CRITIQUE OUR PAPER. THANKS TO YOU, IT IS MUCH IMPROVED!

Reviewer 3 Report

Comments and Suggestions for Authors

Congratulations to the authors. This review on comprehensive biosecurity practices to manage PRRSV is well done and serves as an educational tool for best practices. The authors explain bio-exclusion practices and routes of transmission very well. In section 1, they also mention bio-containment and bio-management, but these subjects are not clearly reviewed and perhaps not the focus of the review, but I suggest they could cite a few relevant references related to bio-containment and bio-management practices for readers in section 1 on page 2 in the paragraphs about bio-containment and bio-management training programs. A suggestion and not a requirement, but could provide some additional references for the readers related to these important topics. 

Some minor line specific revisions:

Line 34: such should be "such as"

Line 52: review should be "review of"

Line 220: of needle-free syringe should be "of a needle-free syringe"

220: what is com? do you mean corn?

230-231: incomplete sentence. Please clarify or add "was done" at end of sentence

231: Start sentence as "Viable PRRSV ...

Author Response

MS 3154156  Response to Reviewers

Greetings. I sincerely would like to thank the review team for helping us to improve our

manuscript. We are excited about this manuscript as it is planned to be the lead off article in

the final publishing of the Special Issue on Biosecuring Animal Populations".

We have attempted to address the points indicated by all three reviewers, and the changes we

made are highlighted in red in the revised manuscript. Specifically, to each point:

Reviewer #3: Comments and Suggestions for Authors

Congratulations to the authors. This review on comprehensive biosecurity practices to

manage PRRSV is well done and serves as an educational tool for best practices. The authors

explain bio-exclusion practices and routes of transmission very well.

THANK YOU VERY MUCH FOR THE COMPLEMENT. IT WAS VERY ENCOURAGING.

In section 1, they also mention bio-containment and bio-management, but these subjects are

not clearly reviewed and perhaps not the focus of the review, but I suggest they could cite a

few relevant references related to bio-containment and bio-management practices for readers

in section 1 on page 2 in the paragraphs about bio-containment and bio-management training

programs. A suggestion and not a requirement, but could provide some additional references

for the readers related to these important topics.

THANK YOU FOR YOUR SUGGESTIONS. ADDITIONAL REFERENCES HAVE BEEN

CITED FOR BIO-CONTAINMENT (REF#3) AND BIO-MANAGEMENT (REF#4, 5).

THEY ARE HIGHLIGHTEND IN RED IN THE REVISED MANUSCRIPT.

Some minor line specific revisions:

Line 34: such should be "such as"

Line 52: review should be "review of"

Line 220: of needle-free syringe should be "of a needle-free syringe"

220: what is com? do you mean corn?

230-231: incomplete sentence. Please clarify or add "was done" at end of sentence

231: Start sentence as "Viable PRRSV ...

THANK YOU FOR POINTING THEM OUT. WE HAVE REVISED EVERY POINTS AND

THEY ARE HIGHLIGHTED IN RED IN THE MANUSCRIPT.

Round 2

Reviewer 2 Report

Comments and Suggestions for Authors
  • A brief summary 

The topic discussed in the paper is certainly of great interest for the pig sector. This is because it focuses on the subject of animal health, but with the latest views on economic and environmental issues as well.

  • General concept comments

The main criticism of this work is that most of the text is devoted to dealing with reviews of factors already well known in the bibliography, and the topic of Next Genaration Biosecurity is now better dealt with but remains in a very descriptive form. It is not clear whether this new proposal only relates to the recategorisation of risk factors or whether there is more to it.

  • Specific comments 

Line 1: specify that it is biosecurity to prevent virus entry

Line 10: it is a problem related to the self-supply of proteins for human use, but also drug reduction and environmental sustainability

Line 492: Does ‘<10%’ mean the % of positive Asians or the risk of entry?

Line 508-5011: . In support of this conclusion, this topic of Next Generation Biosecurity will be re-visited later in this Special Issue on Biosecuring Animal Populations, and follow-up data on PRRSV incidence risk from the third year, as well as a cumulative summary of incidence risk over all three years of the project will be provided

Comments on the Quality of English Language

Just minor correction
